# Variations in de novo donor-specific antibody development among HLA-DQ mismatches in kidney transplant recipients

Peenida Skulratanasak, Thidarat Luxsananun, Nuttasith Larpparisuth*,
Nalinee Premasathian, Attapong Vongwiwatana

Division of Nephrology, Department of Medicine, Faculty of Medicine Siriraj Hospital, Mahidol University,
Bangkok, Thailand

* nuttasith.lap@gmail.com

## Abstract

### Background

HLA-DQ antibodies are the most prevalent de novo donor-specific antibodies (dnDSAs) after kidney transplantation (KT). The immunogenicity and impact of each HLA-DQ mismatch on graft outcomes can vary considerably.

### Methods

This retrospective cohort study investigated the prevalence and risk factors for dnDSA development in patients who underwent KT at Siriraj Hospital between 2006 and 2020 and had HLA-DQB1 mismatches. Our center employed a protocol for post-KT dnDSA surveillance. The impact of dnDSAs on late rejection and graft survival was evaluated.

### Results

In our cohort of 491 KT recipients, 59 (12.02%) developed dnDSAs to HLA-DQB1 at a median time of 4.2 years after KT. The risk of dnDSA occurrence was significantly higher among recipients with HLA-DQ7 mismatch (HR: 2.8; 95% CI: 1.21–6.52; $P$ = 0.017) and HLA-DQ9 mismatch (HR: 2.63; 95% CI: 1.11–6.27; $P$ = 0.028). Recipients who developed dnDSAs were younger ($P$ = 0.009), had higher rates of medication nonadherence ($P$ = 0.031), had pre-KT panel reactive antibody levels above 20% ($P$ = 0.044), and received non-tacrolimus immunosuppression ($P$ < 0.001) compared to those without. Recipients who developed dnDSAs to HLA-DQ exhibited a significantly higher incidence of late graft rejection (HR: 7.76; 95% CI: 5–12.03; $P$ < 0.0001) and inferior death-censored graft survival than those without dnDSAs (log rank $P$ < 0.001).

### Conclusion

The patients with HLA-DQ7 and HLA-DQ9 mismatches exhibit the highest risk of developing dnDSAs. Individualized immunosuppression adjustment and kidney allocation based on specific HLA-DQ mismatch may enhance long-term graft survival.

**Data availability statement:** All relevant data are within the manuscript. The datasets that support the findings of this study are attached along with manuscript.

**Funding:** The author(s) received no specific funding for this work.

**Competing interests:** The authors have declared that no competing interests exist.

## Introduction

Kidney transplantation (KT) offers superior survival and quality of life to other kidney replacement therapies. However, antibody-mediated rejection (AMR) remains a primary cause of late allograft failure, even with advancements in immunosuppressive therapy. KT recipients who are medication nonadherent or underimmunosuppressed may be at higher risk of developing de novo donor-specific antibodies (dnDSAs) and subsequent late AMR, particularly in cases of high numbers of human leukocyte antigen (HLA) mismatches (MM) [1]. Greater compatibility of HLA between donors and recipients is associated with better outcomes in KT.

The most prevalent type of dnDSA detected in cases of late AMR is the anti-HLA-DQB1 antibody. The presence of dnDSAs targeting HLA-DQB1 is associated with the lowest graft survival compared to other types of dnDSA [2,3]. Several studies have reported a significant increase in the risk of graft loss for both living and deceased donor KTs with HLA-DQ MM [4,5]. Moreover, numerous studies have demonstrated a strong association between HLA-DQ MMs and late AMR, leading to subsequently graft failure [5–7]. These findings underscore the high immunogenicity of HLA-DQ over the long term. However, it is noteworthy that most countries do not include HLA-DQ MM in their kidney allocation systems.

Previous studies investigating the influence of HLA/epitope MM on transplant outcomes have predominantly focused on the number of MM. However, each HLA-DQ has demonstrated varying immunogenicity in triggering antibody production and subsequent long-term graft outcomes. A study on heart and lung transplant recipients in Canada found that the epitope MM identified in donor HLA-DQB1*02/HLA-DQB1*03:01 was associated with a higher risk of dnDSA development [8]. We observed an increased frequency of dnDSA against certain HLA-DQ MM compared to others in our center, which has conducted posttransplant dnDSA surveillance for an extended period. Nevertheless, the immunogenic differences among HLA-DQ MMs in KT recipients still require full understanding. Therefore, we conducted this study to identify the specific HLA-DQ MMs associated with dnDSA occurrence and their impact on graft survival. Understanding the differing immunoreactivities among HLA-DQB MM could assist in developing individualized immunosuppression and DSA monitoring for KT recipients.

## Materials and methods

A retrospective cohort study was conducted among adult patients who underwent KT at our center between January 2006 and December 2020. We included recipients with at least one HLA-DQB1 MM. Patients were excluded if they underwent repeat KT, combined transplantation, had pretransplant donor-specific antibodies, experienced primary allograft failure, or lacked post-transplantation data on HLA antibody results. The study was approved by the Ethics Committee of the Faculty of Medicine Siriraj Hospital, Mahidol University, Bangkok, Thailand (COA No. Si 791/2021), with a waiver of informed consent. The study was performed in accordance with international guidelines for human research protection. Patient identifiers were used for data collection but then de-identified after completion of the collection.

Data were obtained from our institutional database and included baseline demographic information, donor profiles, donor and recipient HLA typing, immunosuppressive regimens, self-reported medical nonadherence, development of dnDSAs, and episodes of graft rejections. The database was accessed between November 15, 2021 and March 15, 2022. Depending on the immunologic risk, induction with basiliximab or antithymocyte globulin was prescribed. However, prior to 2012, most patients did not receive antibody induction due to reimbursement limitations. The initial immunosuppressive regimen at our center consisted

of mycophenolic acid, prednisolone, and either cyclosporine or tacrolimus during the earlier period, with tacrolimus becoming the standard during the later phase. Calcineurin inhibitors (CNI) or mycophenolate mofetil were substituted with mTOR inhibitors in cases of infections (e.g., BK viruria/viremia, persistent CMV viremia, or mycobacterial infections), CNI nephrotoxicity, or malignancy. Changes to immunosuppressive regimens were made based on the clinical judgment of the treating physician.

In our center, HLA typing was assessed using intermediate-resolution molecular methods before 2014 and high-resolution typing thereafter. We conducted surveillance for dnDSA at 6 months and 1 year after KT and annually thereafter. HLA antibodies were evaluated using the solid-phase method with a Luminex microbead assay (One Lambda Inc, Canoga Park, CA, USA). The presence of dnDSAs was determined at the antigen level and confirmed using the Luminex LABScreen single antigen method with a mean fluorescence intensity (MFI) cutoff of 1000. The maximal MFI was determined from the bead with the highest MFI in cases where multiple beads for the same antigen were positive. While protocol biopsy is not part of our routine practice, allograft biopsies were performed in KT recipients who experienced unexplained increases in serum creatinine from baseline, persistent proteinuria, or persistently elevated MFI of dnDSA > 5,000 despite adjustments to immunosuppression. Diagnosis of AMR was determined by reviewing pathological scores, DSA results, and C4d staining in accordance with the Banff classification 2019 criteria [9].

The primary outcome was to compare the prevalence of dnDSAs against each HLA-DQ. We defined HLA antibodies as dnDSAs if no such antibodies were detected during the pretransplant or early post-KT period (< 6 months). The prevalence of dnDSAs against each HLA-DQ was determined based on the number of MM. In cases where patients had duplicate DQ MM (e.g., both donor HLA-DQ antigens are DQ5, which mismatches the recipient), such MM were counted as a single dnDSA. The risk factors for dnDSA development and transplant outcomes of KT recipients with dnDSAs against HLA-DQ, including late rejection and allograft failure, were also of interest. Transplant outcomes were assessed at the end of December 2021.

## Statistical analysis

Baseline characteristics are presented as percentages (%), numbers and percentages, or means ± standard deviations for normally distributed data and as medians and interquartile ranges (IQRs) for nonnormally distributed data. Enumerated variables were compared using Chi-squared or Fisher's exact tests, while continuous variables were compared using one-way ANOVA or Kruskal–Wallis H tests, depending on the data distribution. Two-tailed probability ($P$) values < 0.05 were considered statistically significant.

Cox proportional-hazard regression analyses adjusted for mode of KT, donor age, sex, HLA antigen, diabetes, hypertension, death from cerebrovascular accident (CVA), recipient age, sex, diabetes, duration of dialysis, panel reactive antibody (PRA), recipient HLA antigen, other HLA MM, cold ischemic time, and induction and maintenance immunosuppression, were employed to identify factors associated with dnDSA development. Additionally, a Cox model was applied to assess graft outcomes, incorporating the same covariates along with the presence of anti-DQ DSA and class I DSA. The results are reported as hazard ratios (HRs) with 95% confidence intervals. Kaplan–Meier analysis was used to compare the probability of allograft failure after KT. All statistical analyses were performed using IBM SPSS Statistics for Windows, version 21.0 (IBM Corp, Armonk, NY, USA).

## Results

Out of 978 adult patients who underwent KT between January 2006 and December 2020, 451 recipients had no HLA-DQB1 MM. Among the eligible 527 recipients, 17 were excluded

due to repeated KT, 13 due to the presence of preformed donor-specific antibodies, 4 due to combined transplantation, and 2 due to primary allograft failure. Consequently, the analysis encompassed 491 KT recipients with a median follow-up period of 6.35 years (IQR: 3.54–10.22) (Figure 1). All KT had negative complement-dependent cytotoxicity (CDC) and antihuman globulin (AHG) crossmatches. Virtual crossmatch was also performed in patients with high PRA levels, most of whom had single antigen bead testing results. All KT had negative complement dependent cytotoxicity (CDC), antihuman globulin (AHG) and virtual crossmatch. Among them, 75 recipients had 2 HLA-DQB1 MM, bringing the total number of DQB1 MM in this study to 566. Details of the distribution of HLA-DQB1 MM are provided in S1 Table. The mean age of recipients was 41.3 ± 12 years, and 295 patients (60.1%) were male. The majority (96.1%) had pre-transplant PRA levels of < 20%. Table 1 details the baseline characteristics of the enrolled patients stratified by the presence of dnDSAs to HLA-DQB1, irrespective of the development of HLA-nonDQ antibodies. The presence of dnDSAs against HLA-DQB1 was detected in 59 patients (12%) at a median of 4.23 years (IQR: 1.97–5.56) after KT. The mean MFI of the first maximum dnDSA was 9,274 ± 6,405. The incidence of HLA-DQ DSA development was 11.1% and 17.3% in patients with single and double MM, respectively. When compared to patients without anti-DQ dnDSAs, recipients with anti-DQ dnDSAs were significantly younger (35.1 ± 13 vs 42.14 ± 12 years; *P* < 0.001), had PRA class II levels ≥ 20% (10.16% vs 3%; *P* = 0.01), received living donor KT (61% vs 47%; *P* = 0.04), had a lower incidence of delayed graft function (27.1% vs 41.9%; *P* = 0.03), did not receive tacrolimus-based immunosuppression either immediately after KT (55.9% vs 79.6%; *P* < 0.001) or at the time of dnDSA detection (45.8% vs 78.7%; *P* < 0.001) and were more medication nonadherent (6.8% vs 1.9%; *P* = 0.04). There was no significant difference in the prevalence of HLA-nonDQ antibody between the presence and absence of anti-DQ antibody.

## Risk factors for dnDSAs against HLA-DQ

Risk factors for developing dnDSAs against HLA-DQ were determined using Cox proportional hazard model. Univariate analysis revealed significant risk factors, including recipient age under 35 years (HR 3.06; 95%CI 1.34–6.99; *p* = 0.008), pre-transplant PRA class II levels of 20% or higher (HR 2.58; 95%CI 1.11–6.01; *p* = 0.028), non-tacrolimus-based

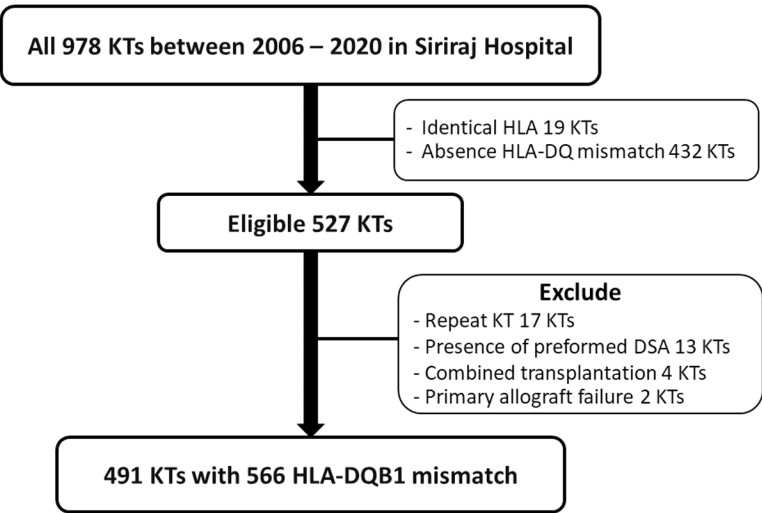

**Fig 1. Consort flow diagram of the study.** KT = kidney transplantation.

**Table 1. Baseline characteristics of participants stratified by presence or absence of anti-HLADQ dnDSA.**

| | Total (N = 491) | Presence of anti-HLADQ dnDSA (n = 59) | No anti-HLADQ dnDSA (n = 432) | P |
|---|---|---|---|---|
| Male gender, n (%) | 295 (60.08%) | 39 (66.10%) | 256 (59.26%) | 0.31 |
| Age, years | 41.30 ± 12 | 35.10 ± 13 | 42.14 ± 12 | **< 0.001** |
| Comorbidities Hypertension, n (%) Diabetes, n (%) | 450 (91.60%) 86 (17.52%) | 52 (88.10%) 8 (13.56%) | 398 (92.10%) 78 (19.9%) | 0.31 0.67 |
| Mode of KRT before KT | | | | |
| Hemodialysis, n (%) | 434 (88.40%) | 51 (86.40%) | 383 (88.60%) | 0.67 |
| Peritoneal dialysis, n (%) | 35 (7.10%) | 4 (6.80%) | 31 (7.20%) | |
| Preemptive KT, n (%) | 22 (4.50%) | 4 (6.80%) | 18 (4.20%) | |
| DDKT, n (%) | 252 (51.32%) | 23 (38.98%) | 229 (53.01%) | **0.04** |
| Donor male gender, n (%) | 267 (54.37%) | 30 (50.84%) | 237 (54.86%) | 0.56 |
| Donor age, years | 38.13 ± 13 | 35.63 ± 13 | 38.47 ± 13 | 0.116 |
| Pre KT PRA class II ≥ 20%, n (%) | 19 (3.88%) | 6 (10.16%) | 13 (3.02%) | **0.01** |
| HLA-DRB1 mismatches | | | | |
| 0, n (%) | 107 (21.79%) | 7 (11.86%) | 100 (23.14%) | 0.14 |
| 1, n (%) | 305 (62.11%) | 41 (69.49%) | 264 (61.11%) | |
| 2, n (%) | 79 (16.08%) | 11 (18.64%) | 68 (15.74%) | |
| HLA-DQB1 mismatches | | | | |
| 1, n (%) | 416 (84.72%) | 46 (77.96%) | 370 (85.64%) | 0.14 |
| 2, n (%) | 75 (15.27%) | 13 (22.04%) | 62 (14.35%) | |
| Delayed graft function, n (%) | 197 (40.12%) | 16 (27.11%) | 181 (41.89%) | **0.03** |
| **Immunosuppression** Antibody induction, n (%) | 290 (59.18%) | 35 (59.32%) | 255 (59.16%) | 0.982 |
| Tacrolimus-use immediate post KT | 377 (76.78%) | 33 (55.93%) | 344 (79.62%) | **< 0.001** |
| Tacrolimus-use at time of dnDSA detection or at time of analysis* | 367 (74.74%) | 27 (45.76%) | 340 (78.70%) | **< 0.001** |
| Immunosuppression at time of dnDSA detection or at time of analysis* Tacrolimus Cyclosporin Mycophenolic acid mTOR inhibitors | 367 (74.75%) 58 (11.81%) 429 (87.37%) 100 (20.37%) | 27 (45.76%) 19 (32.2%) 48 (81.36%) 19 (32.2%) | 340 (78.70%) 39 (9.02%) 381 (88.19%) 81 (18.75%) | **< 0.001** **< 0.001** 0.15 **0.04** |
| History of nonadherence, n (%) | 12 (2.44%) | 4 (6.78%) | 8 (1.85%) | **0.04** |

*Tacrolimus use and immunosuppression regimens were determined at the time of dnDSA detection in patients with dnDSAs and at the time of analysis in the dnDSA-negative group.

Abbreviations: DDKT, deceased donor kidney transplantation; dnDSA, de novo donor-specific antibodies; KRT, kidney replacement therapy; KT, kidney transplantation; PRA, panel reactive antibodies

immunosuppression early after KT (HR 1.79; 95%CI 1.06–3.03; $p$ = 0.03) and at the time of dnDSA detection (HR 1.83; 95%CI 1.42–2.37; $p$ = < 0.001), and medication nonadherence (HR 2.99; 95%CI 1.08–8.25; $p$ = 0.034).

In the multivariate analysis adjusted for crucial factors, recipient age under 35 years (HR 3.16; 95% CI 1.33–7.52; $p$ = 0.009), pre-transplant PRA levels of 20% or higher (HR 2.51; 95% CI 1.03–6.18; $p$ = 0.044), non-tacrolimus-based immunosuppression at the time of dnDSA detection (HR 3.47; 95% CI 1.93–6.25; p < 0.001), and medication non-adherence (HR 3.32; 95% CI 1.12–9.91; $p$ = 0.031) were significantly associated with the development of dnDSAs against HLA-DQ posttransplantation. The statistical results are detailed in Table 2.

**Table 2. Factors associated with development of dnDSA against HLA-DQ in overall cohort.**

| | Univariable analysis | | Multivariable analysis | |
|---|---|---|---|---|
| | HR (95% CI) | *P* | HR (95% CI) | *P* |
| Recipient age (years) > 50 | Reference | | Reference | |
| 35-50 | 1.80 (0.77–4.20) | 0.173 | **2.40 (1.01–5.74)** | **0.047** |
| < 35 | **3.06 (1.34–6.99)** | **0.008** | **3.16 (1.33–7.52)** | **0.009** |
| Recipient sex (male) | 0.90 (0.51–1.59) | 0.712 | | |
| Recipient DM | 0.82 (0.39–1.75) | 0.61 | | |
| Current PRA class II ≥ 20% | **2.58 (1.11–6.01)** | **0.028** | 2.51 (1.03–6.18) | 0.044 |
| DDKT | 0.80 (0.47–1.35) | 0.408 | | |
| Donor age (years) | 0.99 (0.96–1.01) | 0.189 | | |
| Donor sex (male) | 1.12 (0.63–1.97) | 0.702 | | |
| DGF (No DGF) | 1.75 (0.99–3.11) | 0.056 | | |
| Received Ab induction | 1.57 (0.93–2.66) | 0.93 | | |
| Immunosuppression | | | | |
| Non-tac based IS (at transplantation) | **1.79 (1.06–3.03)** | **0.03** | | |
| Non-tac base IS (at dnDSA detection or at analysis$) | **1.83 (1.42–2.37)** | **< 0.001** | **3.47 (1.93–6.25)** | **< 0.001** |
| Nonadherence | **2.99 (1.08–8.25)** | **0.034** | **3.32 (1.12–9.91)** | **0.031** |
| HLA mismatch* | | | | |
| HLA A mismatch | **0.49 (0.26–0.90)** | **0.022** | **0.47 (0.26–0.90)** | **0.015** |
| HLA B mismatch | 1.16 (0.57–2.38) | 0.677 | | |
| HLA DRB1 mismatch | 1.27 (0.52–3.15) | 0.6 | | |
| HLA DQ mismatch* | | | | |
| DQ7 mismatch | 1.7 (0.95–3.05) | 0.076 | **2.80 (1.21–6.52)** | **0.017** |
| DQ8 mismatch | 0.94 (0.4–2.18) | 0.881 | 1.62 (0.62–4.25) | 0.319 |
| DQ9 mismatch | 1.42 (0.74–2.74) | 0.29 | **2.63 (1.11–6.27)** | **0.028** |
| DQ2 mismatch | 1.58 (0.84–2.97) | 0.159 | 2.04 (0.87–4.82) | 0.102 |
| DQ4 mismatch | 1.02 (0.41–2.54) | 0.971 | 2.04 (0.68–6.09) | 0.201 |
| DQ5 mismatch | 0.81 (0.46–1.44) | 0.467 | 2.08 (0.89–4.91) | 0.092 |
| DQ6 mismatch | 0.55 (0.25–1.21) | 0.137 | 0.98 (0.36–2.65) | 0.974 |

*HLA mismatches were analyzed based on all mismatches carried by the recipients.

$Tacrolimus use were determined at the time of dnDSA detection in patients with dnDSAs and at the time of analysis in the dnDSA-negative group.

Abbreviations: DM, diabetes mellitus; DDKT, deceased donor kidney transplantation; DGF, delayed graft function; Ab, antibody; dnDSA, de novo donor specific antibodies; HLA, human leukocyte antigen; IS, immunosuppression; PRA, panel reactive antibodies; tac, tacrolimus

## Specific HLA-DQB1 mismatches and dnDSA development

In overall cohort, the most common specific anti-DQ dnDSA was against HLA-DQ7 (23%), followed by HLA-DQ5 (22%) and HLA-DQ9 (17%) (Fig 2A). The prevalence of dnDSA development per each HLA-DQB1 MM was 11.31% (64/566 MM). The highest prevalence of anti-DQ dnDSA per MM was found in patients who had HLA-DQ7 MM (17.44%), followed by HLA-DQ9 (16.92%) and HLA-DQ2 MMs (14.28%). In contrast, the lowest prevalence was revealed in patients who had HLA-DQ6 MM (5.43%) (Fig 2B).

Among the 15 patients who developed an HLA-DQ7 dnDSA, 7 also exhibited an HLA-DQ7 MM along with another DQ antigen MM. Within this group of 7 patients, 5 presented an obligatory HLA-DQ7 antibody without dnDSAs to another DQ antigen (2 DQ2, 1 DQ8, 1 DQ5, and 1 DQ6), while only 2 patients had dnDSAs to both DQ7 and another

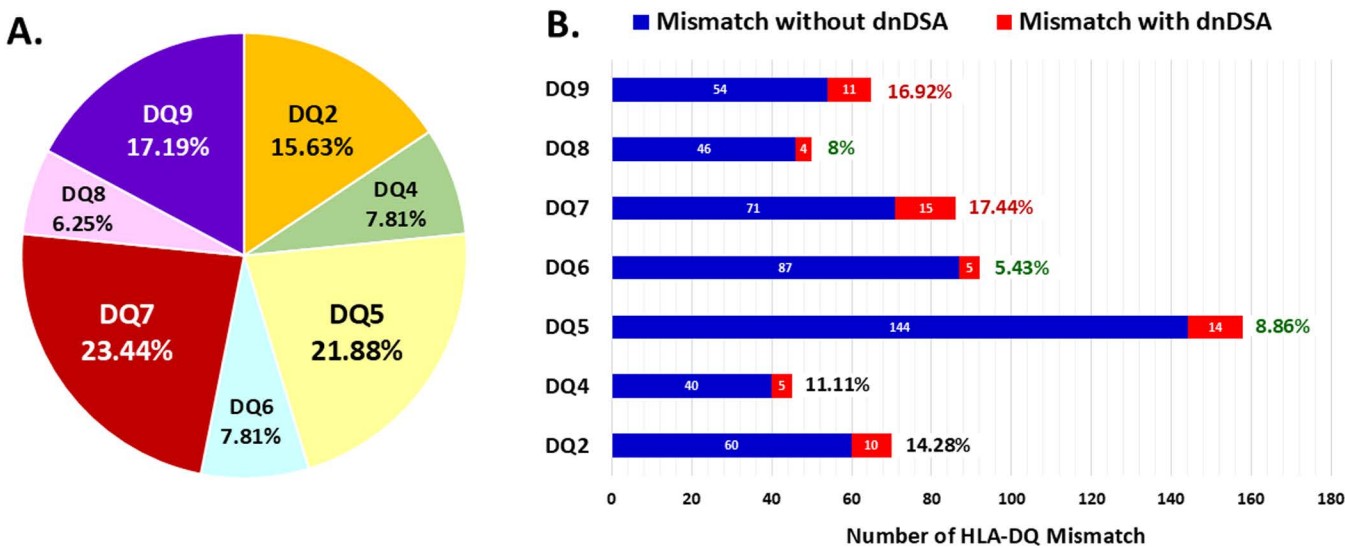

**Fig 2. Type and prevalence of anti-HLA DQ dnDSAs.** A) Type of HLA-DQ dnDSAs found in overall cohort, B) The prevalence of dnDSAs occurrence per each HLA-DQ mismatch in Thai kidney transplant population.

antigen (1 DQ6 and 1 DQ8). In contrast, only 3 out of 70 patients with an obligatory DQ-6 MM developed HLA-DQ6 dnDSA. Out of the 23 patients with an HLA-DQ6 MM along with another DQ antigen MM, 2 developed dnDSAs to DQ6, while 2 had dnDSA occurrences to another DQ (1 DQ7, 1 DQ5) without anti-DQ6 antibody.

We further investigated the association between specific HLA-DQB1 MM and dnDSA development post-KT. Kaplan–Meier analysis revealed that recipients carrying HLA-DQ7 MM had a higher chance of dnDSA occurrence compared to those with HLA-DQ5 (log-rank p = 0.017) and HLA-DQ6 (log-rank p = 0.005), respectively. Recipients with HLA-DQ9 MM faced a significantly higher risk of dnDSA development than those with HLA-DQ6 (log-rank p = 0.026). No statistically significant differences were observed in other DQ antigen comparisons (Fig. 3).

The adjusted Cox model demonstrated that recipients with HLA-DQ7 MM had the highest risk of dnDSA formation (HR: 2.80; 95% CI: 1.21–6.52; $p$ = 0.017), followed independently by those with HLA-DQ9 MM (HR: 2.63; 95% CI: 1.11–6.27; $p$ = 0.028) (Table 2). When specifically compared to recipients with HLA-DQ6 MM, those with HLA-DQ7 MM (HR: 3.67; 95% CI: 1.20–11.25; $p$ = 0.023) and HLA-DQ9 MM (HR: 3.391; 95% CI: 1.08–10.65; $p$ = 0.037) had a significantly higher risk of dnDSA occurrence. No statistically significant differences were observed in other pairwise comparisons of DQ MM.

### De novo anti-DQ DSA, non-DQ DSA and graft outcomes

Of the 491 KT recipients included in the study, 139 underwent allograft biopsy more than 6 months after KT due to allograft dysfunction, presence of dnDSAs, or clinical suspicion of glomerulonephritis. Biopsy-proven late rejection was found in 72 patients. Late active or chronic active AMR was diagnosed in 69.5% of patients with HLA-DQ dnDSAs. Kaplan–Meier analysis revealed that recipients with dnDSAs to HLA-DQ had significantly inferior rejection-free survival compared to those without dnDSAs to HLA-DQ (HR: 7.76; 95% CI: 5.00–12.03; $P$ < 0.0001; Fig 4A). While the 5-year death-censored graft survival was similar for recipients with and without dnDSAs (94.6% vs 93.8%, $P$ = 0.452), the 10-year survival was

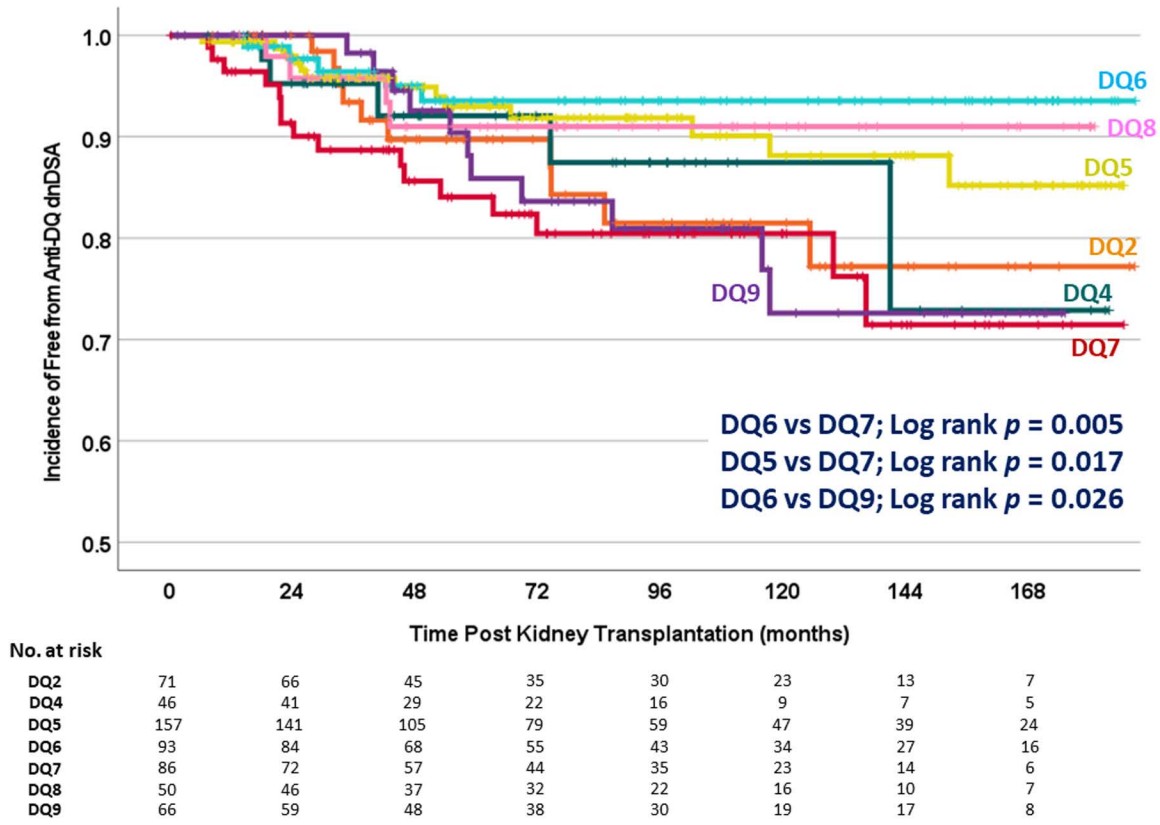

**Fig 3. Kaplan-Meier analysis compared rate of dnDSA occurrence among each HLA-DQ mismatches.**

significantly lower for recipients with dnDSAs to HLA-DQ than for those without dnDSAs (70.2% vs 87.8%; *P* = 0.001). Overall graft survival was also inferior in those who had dnDSA to HLA-DQ (log rank *P* < 0.001; Fig 4B). In an adjusted Cox proportional hazards model, the presence of dnDSAs to HLA-DQ remained significantly associated with overall graft failure (HR: 3.875, 95% CI: 2.12–7.07) (Table 3).

Across the entire cohort, 20 patients (4.1%) had isolated non-DQB1 DSAs without accompanying DQB1 DSAs. Kaplan-Meier analysis revealed no statistically significant difference in rejection-free survival between recipients with HLA-DQ dnDSAs and those with non-DQ DSAs (log-rank P = 0.111) S1 Fig. However, recipients with dnDSAs to HLA-DQ had significantly lower overall graft survival compared to those with non-DQ DSAs (log-rank P = 0.038) S2 Fig. Additionally, survival analysis showed no significant difference in graft survival between recipients with non-DQ DSAs and those without DSAs (log-rank P = 0.67) S3 Fig.

## Discussion

Our study is the first to demonstrate varying immunogenicity among specific HLA-DQ MM in KT recipients. Patients with HLA-DQ7 and HLA-DQ9 MM are at the highest risk for dnDSA occurrence, while those with HLA-DQ6 MM have the lowest risk, even when adjusted for crucial transplant factors. Our findings regarding risk factors for HLA-DQ dnDSA development, such as young age, high pre-KT PRA levels, medication nonadherence, and non-tacrolimus-based immunosuppression, are consistent with previous studies [1,10,11].

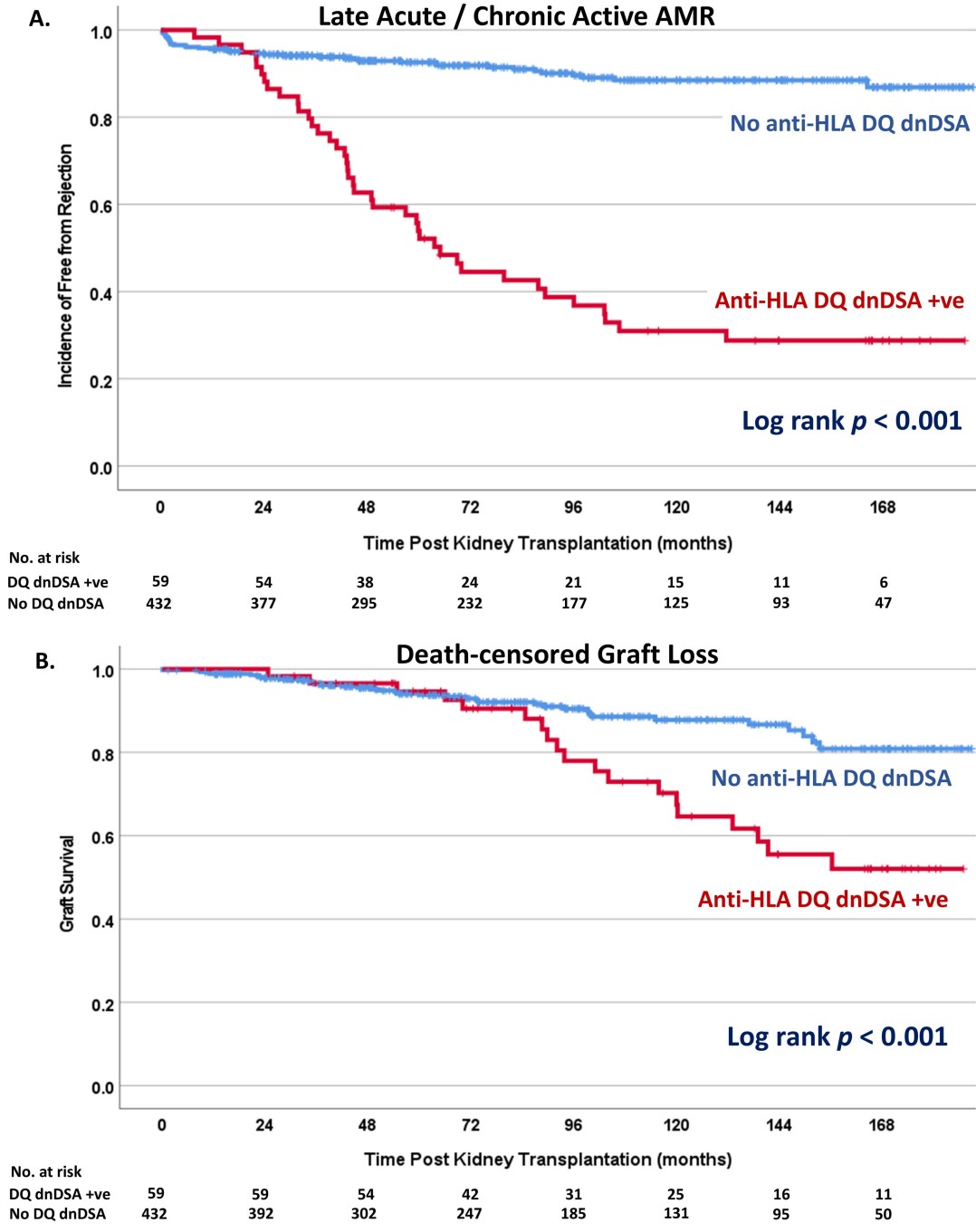

**Fig 4. Kaplan-Meier analysis compared between recipients who had HLA-DQ dnDSA and those without.** A) Biopsy-proven acute or chronic-active rejection, B) Death-censored graft survival.

Our investigation also confirms the significant impact of anti-DQ dnDSA development on long-term adverse outcomes, including late AMR and graft loss [12–14].

Despite advancements in medical care for transplant recipients, allograft rejection remains a major barrier to long-term graft survival. The detrimental effects of dnDSAs are well known, and dnDSAs can lead to late AMR, chronic glomerulopathy, and graft loss. The reported prevalence of DSAs has varied between studies, ranging from approximately 5% to 30% over

**Table 3. Factors associated with graft failure in overall cohort.**

| | Univariable analysis | | Multivariable analysis | |
|---|---|---|---|---|
| | HR (95% CI) | *P* | HR (95% CI) | *P* |
| Recipient age (years)<br>> 50 | Reference | | | |
| 35-50 | 1.15 (0.48–2.79) | 0.751 | | |
| < 35 | 1.11 (0.48–2.55) | 0.806 | | |
| Recipient sex (male) | 1.30 (0.71–2.39) | 0.391 | | |
| Recipient DM | 1.44 (0.53–3.91) | 0.477 | | |
| Current PRA class II ≥ 20% | 1.19 (0.54–2.63) | 0.667 | | |
| DDKT | 1.42 (0.62–3.26) | 0.402 | | |
| Donor age (years) | **1.04 (1.01–1.06)** | **0.003** | **1.04 (1.01–1.06)** | **0.002** |
| Donor sex (male) | 0.77 (0.41–1.48) | 0.439 | | |
| DGF (Presence of DGF) | **2.65 (1.30–5.42)** | **0.003** | **2.61 (1.52–4.49)** | **< 0.001** |
| Received Ab induction | 1.23 (0.65–2.35) | 0.527 | | |
| Immunosuppression | | | | |
| Non-tac based IS<br>(at transplantation) | 0.77 (0.39–1.54) | 0.466 | | |
| Non-tac base IS<br>(at dnDSA detection or at analysis$) | 1.04 (0.40–2.73) | 0.742 | | |
| Nonadherence | **3.17 (1.16–8.66)** | **0.027** | **3.32 (1.12–9.91)** | **0.031** |
| HLA mismatch* | | | | |
| HLA A mismatch | 1.00 (0.49–2.04) | 0.996 | | |
| HLA B mismatch | 1.90 (0.83–4.34) | 0.127 | | |
| HLA DRB1 mismatch | 2.32 (1.01–5.32) | 0.046 | | |
| HLA DQ mismatch* | | | | |
| DQ2 mismatch | 0.65 (0.23–1.85) | 0.415 | | |
| DQ4 mismatch | 1.46 (0.45–4.74) | 0.53 | | |
| DQ5 mismatch | 0.67 (0.25–1.79) | 0.424 | | |
| DQ6 mismatch | 0.93 (0.34–2.55) | 0.885 | | |
| DQ7 mismatch | 0.40 (0.13–1.24) | 0.114 | | |
| DQ8 mismatch | 0.43 (0.11–1.71) | 0.232 | | |
| DQ9 mismatch | 0.36 (0.11–2.74) | 0.085 | | |
| Acute cellular rejection (Y) | 1.04 (0.45–2.34) | 0.655 | | |
| Class I DSA | 0.45 (0.94–2.12) | 0.31 | | |
| Anti-HLA DQ DSA | **3.19 (1.59–6.42)** | **0.001** | **3.88 (2.12–7.07)** | **< 0.001** |

*HLA mismatches were analyzed based on all mismatches carried by the recipients.

$Tacrolimus use were determined at the time of dnDSA detection in patients with dnDSAs and at the time of analysis in the dnDSA-negative group.

Abbreviations: DM, diabetes mellitus; DDKT, deceased donor kidney transplantation; DGF, delayed graft function; Ab, antibody; dnDSA, de novo donor specific antibodies; HLA, human leukocyte antigen; IS, immunosuppression; PRA, panel reactive antibodies; tac, tacrolimus

a 1- to 10-year period after transplantation [15]. The median times at DSA detection post-KT in cohort studies from the United Kingdom and the United States were 9.8 and 19.5 months, respectively, earlier than our findings [12–16]. However, our median time of 4.2 years is comparable to reports from Canada and our antecedent studies [1,17,18].

The explanation for the relatively long time to dnDSA detection post-KT is the differences in immunologic risks and pharmacokinetic profiles of KT recipients. Most recipients in Thailand were nonsensitized and received their first solid organ transplantation, which is associated with a lower incidence of dnDSA development [19]. Our recipients also experienced

higher MPA levels when exposed to a similar MPA dosage as the western population, which may have protected them against DSA occurrence [20,21]. Generally, the occurrence of dnD-SAs precedes late AMR, which may remain asymptomatic for a relatively long period. Therefore, we did not observe a difference in 5-year survival; however, the 10-year graft outcome was markedly and significantly inferior in the dnDSA group.

Our cohort's risk factors associated with anti-DQ dnDSA formation are consistent with previous studies. Younger patients tend to have a more robust immunologic response, resulting in a higher chance of dnDSA occurrence after transplantation. Medication non-adherence is a well-known contributor to under-immunosuppression and the subsequent development of dnDSAs and late rejection [22–24]. Recipients with high pretransplantation PRA levels may have cryptic memory responses to various HLA antigens, leading to early dnDSA development [25]. Tacrolimus-based immunosuppression with archival target trough levels is associated with a lower incidence of dnDSA development [26]. The high rate of rejection and subsequent unfavorable graft outcomes observed in the anti-DQ dnDSA group in our study confirms the impact of dnDSAs observed in antecedent studies [2,12]. Another important finding in our study showed comparable rejection-free survival between recipients with anti-DQ and non-DQ dnDSAs; however, those with anti-DQ DSAs had significantly worse graft survival. This may indicate that anti-DQ DSAs exert a more detrimental effect on long-term graft outcomes compared to other types of DSAs. Further studies are required to elucidate this observation and better understand the clinical relevance of different DSA types.

Anti-HLA-DQB1 is the most common type of dnDSA observed after KT, as seen in studies conducted in Thailand [6,7,18] and other populations globally [3,7,27]. The prevalence of specific HLA-DQB1 dnDSAs can differ among various populations. For example, a South Korean study found that the most frequent anti-DQ dnDSAs were directed against DQ6 (39.4%), followed by DQ9 (27.3%) and DQ8 (24.3%) [2]. However, a US-based study revealed that the most common dnDSAs were directed against DQ7 (25%), DQ2 (19%), and DQ4 (19%) [12]. Nonetheless, these studies did not report the ratio of recipients with dnDSAs to those carrying HLA-DQB1 MM.

The most prevalent anti-DQ dnDSAs found in our cohort were against HLA-DQ7 (24%), HLA-DQ5 (22%) and HLA-DQ9 (18%). However, the most common HLA-DQB1 MM found in recipients are DQ5 (32.2%), DQ6 (18.7%), and DQ7 (17.5%) MM. Therefore, when analyzing the ratio of dnDSA per MM, the most frequently occurring specific DQ antibodies were against DQ7 (17.4%), followed by DQ9 (16.9%) and DQ2 (14.3%). Adjusted Cox model confirmed that recipients carrying HLA-DQ7 and DQ9 MM are at the highest risk for dnDSA development. The genetic polymorphism of each population might influence the differences in the occurrence specific HLA-DQ antibodies after KT.

Although the hypothesis regarding dissimilarity in immunogenicity among each HLA MM in KT is not well established, a study on heart and/or lung transplant recipients from Canada showed that an epitope MM of HLA-DQA1*05+DQB1*02 or DQB1*07 increased the risk for dnDSA occurrence by approximately 4-fold [8]. In general, DQB1 is co-localized with DQA1 antigen as a heterodimer on the surface of leukocytes. Variations in DQA1 typing may impact immunogenicity of the DQB1 antigen. However, the population in our country is homogeneous, primarily comprising Thai ethnics, resulting in less diverse HLA typing. Additionally, DQA1 typing in our center was conducted after 2014. Recipients with DQB1*7 mostly (75%) carried the gene DQA1*0601+DQB1*0301 which is relatively restricted in Southeast Asia and the South Pacific, while the remaining 25% carried the gene DQA1*0505+DQB1*0301, consistent with a previous study [28]. Our findings, in line with previous Canadian data in heart and lung transplantation [8], support that the notion that the problematic MM might be DQ7 more than the colocalized DQA. All recipients who had DQ9 carried the gene

DQA1*0302+DQB1*0303. Therefore, determining pathogenic MM in each population should be based on their own data.

The current concept regarding HLA and epitope MM focuses on the number of MM rather than the specific HLA types [5,29]. Our study also revealed that recipients with double DQ MM tended to be at higher risk for DSA development than those with single MM (17.3% vs. 11.1%). However, the rate of DSA development in double MM was not twice as high as in single MM, suggesting that the quality of MM may play a more meaningful role in DSA formation than the quantity alone. Patients with two low-immunogenicity MM may have a lower risk of developing dnDSA than those with one high-risk MM. Notably, HLA-DQ5 and HLA-DQ6 MM have an even higher epitope load than HLA-DQ7 or HLA-DQ9 [30,31]. Further studies are needed to explore the qualitative differences in epitope immunogenicity, as some epitopes may elicit stronger alloimmune responses than others. Identifying common pathologic HLA antigen or epitope MM in the population could inform personalized immunosuppression and dnDSA surveillance to mitigate high-risk MM-associated dnDSAs. Future allocation systems should consider preventing the generation of high-risk MM rather than solely matching the number of HLA MM.

Our study's strengths include a large population size with an extended follow-up, the availability of HLA-DQB typing information even for earlier patients, routine DSA surveillance, and low loss to follow-up (approximately 1.2%). However, several limitations should be acknowledged. First, data on HLA-DQA1 typing were unavailable for some earlier patients, however the association between HLA-DQB1 and HLA-DQA1 is relatively restricted in our population. Second, protocol biopsy was not performed routinely, and graft biopsy was not performed in all cases with dnDSAs, as some cases resolved after adjusting immunosuppression. Therefore, early AMR may have been underdiagnosed, particularly in cases with low DSA MFI. Third, incomplete data on BK nephropathy in our cohort, prevented us from including this factor in the survival analysis. BK nephropathy is a known contributor to graft loss, and future studies with complete data on this variable would provide a more comprehensive understanding of the impact of dnDSA on graft outcomes. Lastly, epitope MM assessment was unfeasible for earlier patients without complete HLA typing. The number and rate of interesting outcomes among patients who underwent higher-resolution HLA typing lacked sufficient power to evaluate the effect of epitope MM. Therefore, we plan to investigate this issue in future studies. However, currently, the detection of dnDSA still relies on HLA antigen rather than epitope, and assessing antigen MM remains more applicable in several countries where high-resolution HLA typing is not routinely performed due to higher associated costs.

## Conclusion

Our findings support the notion that anti-HLA-DQB1 antibodies are significantly associated with allograft rejection and inferior graft outcome. Moreover, our study demonstrated differences in the immunogenicity of HLA-DQ MM in KT recipients, with patients who carry HLA-DQ7 and HLA-DQ9 MM having a significantly higher risk of dnDSA occurrence than other DQ MM. Based on these results, it may be worthwhile to consider kidney allocation and individualized adjustment of immunosuppression based on the type of HLA-DQ MM to prevent dnDSA development and improve long-term graft survival.

## Supporting information

**S1 Table. Distribution of HLA-DQB1 MM in overall cohort.**
(DOCX)

**S1 Fig. Kaplan-Meier analysis compared rejection-free survival between recipients with HLA-DQ dnDSAs and those with non-DQ DSAs.**
(TIF)

**S2 Fig. Kaplan-Meier analysis compared death-censored graft survival between recipients with HLA-DQ dnDSAs and those with non-DQ DSAs.**
(TIF)

**S3 Fig. Kaplan-Meier analysis compared death-censored graft survival between recipients with non-DQ DSAs and those without DSAs.**
(TIF)

## Acknowledgements

The authors thank Mr. David Park for his assistance with language editing.

## Author contributions

**Conceptualization:** Peenida Skulratanasak, Thidarat Luxsananun, Nuttasith Larpparisuth, Nalinee Premasathian, Attapong Vongwiwatana.

**Data curation:** Peenida Skulratanasak, Thidarat Luxsananun, Nuttasith Larpparisuth, Attapong Vongwiwatana.

**Formal analysis:** Peenida Skulratanasak, Thidarat Luxsananun, Nuttasith Larpparisuth, Nalinee Premasathian, Attapong Vongwiwatana.

**Investigation:** Thidarat Luxsananun, Nuttasith Larpparisuth, Nalinee Premasathian.

**Methodology:** Peenida Skulratanasak, Thidarat Luxsananun, Nuttasith Larpparisuth, Attapong Vongwiwatana.

**Project administration:** Nuttasith Larpparisuth.

**Resources:** Nuttasith Larpparisuth.

**Software:** Nuttasith Larpparisuth.

**Supervision:** Peenida Skulratanasak, Nuttasith Larpparisuth, Nalinee Premasathian, Attapong Vongwiwatana.

**Validation:** Peenida Skulratanasak, Nuttasith Larpparisuth.

**Writing – original draft:** Peenida Skulratanasak, Thidarat Luxsananun, Nuttasith Larpparisuth, Nalinee Premasathian, Attapong Vongwiwatana.

**Writing – review & editing:** Peenida Skulratanasak, Nuttasith Larpparisuth, Nalinee Premasathian, Attapong Vongwiwatana.

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
