## [Decision Letter · Decision Letter 0]

4 Dec 2024

PONE-D-24-21041Variations in De Novo Donor-Specific Antibody Development Among HLA-DQ Mismatches in Kidney Transplant RecipientsPLOS ONE

Dear Dr. Larpparisuth,

Thank you for submitting your manuscript to PLOS ONE. After careful consideration, we feel that it has merit but does not fully meet PLOS ONE’s publication criteria as it currently stands. Therefore, we invite you to submit a revised version of the manuscript that addresses the points raised during the review process.

The manuscript has been evaluated by three reviewers, and their comments are available below.

The reviewers have raised a number of concerns that need attention. They request additional information on methodological aspects of the study, further clarification of the results and additional discussion of potential clinical relevance.

Could you please revise the manuscript to carefully address the concerns raised?==============================

We look forward to receiving your revised manuscript.

Kind regards,

Johanna Pruller, Ph.D.

Associate Editor

PLOS ONE

Journal Requirements:

2. In the online submission form, you indicated that all relevant data are within the manuscript. The datasets that support the findings of this study are available from the corresponding author upon reasonable request. 

3. Please remove your figures from within your manuscript file, leaving only the individual TIFF/EPS image files, uploaded separately. These will be automatically included in the reviewers’ PDF.

Reviewers' comments:

Reviewer's Responses to Questions

**Comments to the Author**

1. Is the manuscript technically sound, and do the data support the conclusions?

Reviewer #1: Partly

Reviewer #2: Partly

Reviewer #3: Yes

2. Has the statistical analysis been performed appropriately and rigorously? 

Reviewer #1: Yes

Reviewer #2: Yes

Reviewer #3: Yes

3. Have the authors made all data underlying the findings in their manuscript fully available?

Reviewer #1: Yes

Reviewer #2: Yes

Reviewer #3: Yes

4. Is the manuscript presented in an intelligible fashion and written in standard English?

Reviewer #1: Yes

Reviewer #2: Yes

Reviewer #3: Yes

5. Review Comments to the Author

Reviewer #1: This is a single-centre, retrospective study conducted on a cohort of kidney transplant and SPK first-graft recipients over a 14yrs period aimed to identify the specific HLA-DQB1 mismatches associated with de novo DSA occurrence and their impact on graft survival. Authors found that patients with HLA-DQ7 and HLA-DQ9 MMs had the highest risk of developing de novo DSAs, and that DSAs against DQB1 MM were associated with detrimental graft outcomes.

It is interesting to see that authors have interestingly managed to cast some light on the importance of matching against DQB1 antigens, raising concern about the higher risk of developing DSAs against DQ7 and DQ9. The association between DSAs against DQB1 antigens and rejection / allograft loss is not a new finding and has extensively been described in the literature (of note - the works from Aleksandar Senev and co-workers should be cited as remarkable studies highlighting the role of DSAs against DQs and poor transplant outcomes). Therefore, the authors should revisit the survival analysis and address the followings:

- Is the survival analysis the result of a crude analysis? Although well known from the literature, it is difficult to state that DSAs against DQB1 MMs are associated with detrimental outcomes in the in-study cohort if the survival analysis doesn't consist in a cox-regression taking into account

1) HLA-A / HLA-B MM (and possibly HLA-Cw where available)

2) co-existence of class I and class II DSAs

3) prevalence of cellular rejection and BK nephropathy

- was the incidence of AMR under-estimated? according to the methods, a biopsy was done only if DSA MFI > 5,000 or change in function - early AMR stage may have been missed

- did the authors look into the selective pathogenicity of DSAs against specific DQs?

Minor revision: page 7 line 112: the term 'repeated DQ MM' is inappropriately used in my opinion, as it should instead refer to MMs which recurs in recipients of >1 graft.

Reviewer #2: In their manuscript "Variations in De Novo Donor-Specific Antibody Development Among HLA-DQ Mismatches in Kidney Transplant Recipients" Skulratanasak and colleagues retrospectively investigated a cohort of 491 kidney transplant recipients with HLA-DQB1 mismatches. They analyzed individually DQ loci carried by the recipient, and the likelihood of developing donor specific antibodies (DSAs). They found that patients who developed DSAs to HLA-DQ had increased risk of rejection and graft loss, in line with previously published results. They found that some recipient antigen mismatches such as DQ7 and DQ9 are at highest risk of inducing DSAs. The methodology used is mostly appropriate. The manuscript is sometimes difficult to follow. While there is some originality to the main finding of the manuscript, its clinical relevance needs to be further clarified. Specifically, it remains to be demonstrated whether the stratification in risk of developing DSAs (figure 3) also correlates with stratified rejection-free survivals and graft outcomes. I would encourage the authors to consider the following additional issues should they consider resubmitting the manuscript to a different journal:

- Data on DR mismatches should be provided. In its current form, it is impossible to determine whether clinical outcomes were influenced by DR mismatches;

- Prior studies have demonstrated that the number of DQ mismatches influences the development of DSAs. Here, the authors have focused on specific allele mismatches and development of DSAs. What was the distribution of single vs double mismatches in for each allele? In other words, were there allele mismatches that were more likely to be found in isolation and allele mismatches that were more likely to be found in a double mismatch? This could also influence differences in development of DSAs.

- How was AMR defined?

- Was there a specific protocol for induction immunosuppression, or was this left to the discretion of the treating provider? Similarly, can you provide additional information on how maintenance immunosuppression was decided?

- The tables, their abbreviations/notes and the figure legends are embedded in the main text and this makes following the text more difficult. I would recommend placing the tables and figure legends at the end of the manuscript.

- In the methods you mention that "Simultaneous kidney-pancreas patients should be removed from the analysis, given the different potential for Ab development and rejection" - if SPK patients were indeed included, I would recommend removing them from the analysis

- In the introduction, it is stated that "compatibility of HLA between donors and recipients is crucial in achieving successful KT" - I would revise this sentence, as many recipients achieve excellent outcomes despite receiving fully mismatched kidneys

- Line 199: "The prevalence of dnDSA development per each HLA-DQB1 MM was 11.25% (64/569 MM)." - the number of mismatches should be 566, consistently with the rest of the manuscript

- Table 1: "Immunosuppression at time of dnDSA detection" by definition cannot be applied to the "no DSA" column. So what was the time point used to assess the immunosuppression reported?

Reviewer #3: The manuscript describes a single-center, retrospective, cohort study in 491 kidney transplant recipients where they sought to identify the HLA-DQ mismatches with the highest risk of developing de novo DSA. Patients included in this study have at least one HLA-DQ mismatch, have not received a previous kidney transplant, and did not show DSA >1000 MFI pre-transplant. The authors found that 12% developed de novo DSA to HLA-DQ mismatches and multivariate analysis showed that those with DQ7 and DQ9 mismatches carried an increased risk for developing de novo DSA, despite not being the most found DQ mismatches. Finally, they show that patients who developed de novo DQ DSA, compared to patients without de novo DQ DSA, had an increased incidence of late active or chronic AMR and lower 10-year death-censored graft survival. This is a well-written paper on a currently popular topic. This manuscript adds to the existing knowledge about HLA-DQ mismatches and de novo DSA formation and its clinical relevance in transplantation.

Comments:

1. Materials and Methods section, page 6, starting in line 96: Please describe more extensively how the assignment of HLA-DQ DSA was done

1.a. Was DSA assigned at the antigen or allele level (manuscripts state that HLA typing was performed by intermediate or high resolution, depending on whether it was before vs after 2014)

1.b. When you assigned DSA by antigen, how did you handle cases where multiple beads with the same antigen, but different alleles were above your cutoff?

1.c. DQA DSA independent from DQB DSA or DQA-DQB DSA as heterodimer

2. Results section,

2.a. Page 8, line 142: Please briefly describe how the virtual XM was done.

2.b. Page 12

2.b.i. Line 197: “the most common specific anti-DQ dnDSA was against HLA-DQ7 (24%)”, however, figure 2A shows 23.44% which rounded to the nearest whole number is 23%. Please clarify.

2.b.ii. Line 198: 18% for DQ9 but figure 2A shows 17.19% which rounded to the nearest whole number is 17%. Please clarify.

2.b.iii. Line 200: The prevalence of DQ dnDSA per DQ antigen says DQ9 is 16.92% but figure 2B shows 16.67%, DQ2 says 12.85% but the figure shows 14.08%. Please clarify.

2.c. Page 13

2.c.i. Line 215-220 is confusing. Why are DQ7 mismatches compared to DQ5 and DQ6, and DQ9 compared to DQ6?

2.c.ii. Line 226-228, like 4.a., why are DQ7 and DQ9 compared to DQ6?

2.c.iii. Line 234 and figure 4A shows that patients with dnDSA to DQ have lower rejection-free survival compared to patients with no dnDSA to DQ.

2.c.iii.1. Were there any differences between different DQ DSA (i.e., DQ7 or DQ9 DSA worse than other DQ DSA?

2.c.iii.2. How about DQ DSA versus non-DQ DSA? In other words, is it clinically relevant that certain DQ mismatches seem to be more “immunogenic”?

3. Discussion section

3.a. Page 16, line 295: When the authors refer to ratio of dnDSA per MM, shouldn’t this use the rank and percentages on figure 2B rather than 2A?

3.b. Page 16, line 303, the study cited is from Canada and not from the US

3.c. Page 7, line 313, the study is from Canada and not from the US

6. PLOS authors have the option to publish the peer review history of their article (what does this mean? ). If published, this will include your full peer review and any attached files.

**Do you want your identity to be public for this peer review?** For information about this choice, including consent withdrawal, please see our Privacy Policy .

Reviewer #1: No

Reviewer #2: No

Reviewer #3: **Yes: ** Hugo Kaneku

---

## [Author Response · Author response to Decision Letter 1]

13 Jan 2025

Journal Requirements

In the online submission form, you indicated that all relevant data are within the manuscript. The datasets that support the findings of this study are available from the corresponding author upon reasonable request.

Responses:

We agree with journal requirement. We have already attached our dataset regarding this study in this manuscript, as your recommendations. For rebuttal, we have already addressed all comments from reviewers with a point-by-point response, as recommended in the guidelines.

From Reviewer #1,

This is a single-centre, retrospective study conducted on a cohort of kidney transplant and SPK first-graft recipients over a 14yrs period aimed to identify the specific HLA-DQB1 mismatches associated with de novo DSA occurrence and their impact on graft survival. Authors found that patients with HLA-DQ7 and HLA-DQ9 MMs had the highest risk of developing de novo DSAs, and that DSAs against DQB1 MM were associated with detrimental graft outcomes.

It is interesting to see that authors have interestingly managed to cast some light on the importance of matching against DQB1 antigens, raising concern about the higher risk of developing DSAs against DQ7 and DQ9. The association between DSAs against DQB1 antigens and rejection / allograft loss is not a new finding and has extensively been described in the literature (of note - the works from Aleksandar Senev and co-workers should be cited as remarkable studies highlighting the role of DSAs against DQs and poor transplant outcomes).

Responses:

We sincerely thank the reviewer for the thoughtful and constructive feedback on our manuscript. We agree that the association between DSAs against HLA-DQB1 mismatches and adverse transplant outcomes has been previously described in the literature. However, we believe our study provides additional insights by highlighting the specific risk associated with HLA-DQ7 and HLA-DQ9 mismatches, which has not been extensively explored.

We appreciate your suggestion to cite the important work of Senev A et al., which has significantly contributed to the understanding of HLA mismatches and their impact on transplant outcomes. We have now cited the study by Senev A et al. in our manuscript:

Senev A, Coemans M, Lerut E, Van Sandt V, Kerkhofs J, Daniëls L, et al. Eplet Mismatch Load and De Novo Occurrence of Donor-Specific Anti-HLA Antibodies, Rejection, and Graft Failure after Kidney Transplantation: An Observational Cohort Study. J Am Soc Nephrol. 2020;31(9):2193-2204.

This citation has been added as reference number 5 to support the significance of HLA-DQ mismatches in the development of dnDSAs and their impact on transplant outcomes.

1. Is the survival analysis the result of a crude analysis? Although well known from the literature, it is difficult to state that DSAs against DQB1 MMs are associated with detrimental outcomes in the in-study cohort if the survival analysis doesn't consist in a cox-regression taking into account

1) HLA-A / HLA-B MM (and possibly HLA-Cw where available)

2) Co-existence of class I and class II DSAs

3) Prevalence of cellular rejection and BK nephropathy

Responses

Thank you for your thoughtful comment regarding our survival analysis. We initially reported crude survival analysis in the manuscript, showing that dnDSAs to HLA-DQ were significantly associated with inferior long-term graft survival. In response to your suggestion, we have now performed a Cox proportional hazards regression model to account for potential confounding factors, including HLA-A, HLA-B mismatches, the presence of class I and class II DSAs, and acute cellular rejection.

Our adjusted analysis confirmed that dnDSAs to HLA-DQ were significantly associated with overall graft failure (HR: 3.875, 95% CI: 2.12–7.07). These findings strengthen the association between dnDSAs to HLA-DQ and detrimental graft outcomes in our cohort.

Updated text in manuscript (Results Section):

"While the 5-year death-censored graft survival was similar for recipients with and without dnDSAs (94.6% vs. 93.8%, P = 0.452), the 10-year survival was significantly lower for recipients with dnDSAs to HLA-DQ compared to those without dnDSAs (70.2% vs. 87.8%; P = 0.001). Overall graft survival was also inferior in recipients with dnDSAs to HLA-DQ (log-rank P < 0.001; Fig. 4B). In an adjusted Cox proportional hazards model, the presence of dnDSAs to HLA-DQ remained significantly associated with overall graft failure (HR: 3.875, 95% CI: 2.12–7.07) (Table 3)."

We acknowledge that we have limited data on BK nephropathy in our cohort, and therefore this factor was not included in the Cox regression model. We have added this as a limitation in the revised manuscript.

Updated text in manuscript (Discussion Section):

"Incomplete data on BK nephropathy in our cohort, prevented us from including this factor in the survival analysis. BK nephropathy is a known contributor to graft loss, and future studies with complete data on this variable would provide a more comprehensive understanding of the impact of dnDSA on graft outcomes. "

Thank you for your valuable suggestion, which has helped improve the depth and rigor of our analysis. We believe that the detrimental outcomes associated with anti-DQ antibodies observed in our study are due to the long follow-up period.

2. Was the incidence of AMR under-estimated? according to the methods, a biopsy was done only if DSA MFI > 5,000 or change in function - early AMR stage may have been missed.

Responses

We appreciate your insightful comment regarding the potential underestimation of antibody-mediated rejection (AMR). We acknowledge that early AMR could have been missed, particularly in cases where donor-specific antibodies (DSAs) had a mean fluorescence intensity (MFI) below the threshold for performing a biopsy. In response to your suggestion, we have revised the limitations section of our manuscript to reflect this concern. The statement now reads:

“Second, protocol biopsy was not performed routinely, and graft biopsy was not performed in all cases with dnDSAs, as some cases resolved after adjusting immunosuppression. Therefore, early AMR may have been underdiagnosed, particularly in cases with low DSA MFI.”

This revision highlights the possibility of missed early AMR cases and addresses the limitations in our study design. We believe this addition will improve the clarity and transparency of our manuscript.

3. Did the authors look into the selective pathogenicity of DSAs against specific DQs?

Responses

We appreciate your perceptive comment. We have considered the selective pathogenicity of DSAs against specific HLA-DQ antigens, particularly the higher risk associated with HLA-DQ7 and HLA-DQ9 mismatches. Interestingly, a previous study in heart transplantation also reported an association between HLA-DQ7 and de novo DSA development, which supports our findings.

As discussed in our manuscript, we analyzed the epitope load for each HLA-DQ mismatch and found that the epitope load for DQ7 and DQ9 is lower than that of other DQ types, such as DQ6, which exhibited a lower incidence of DSA formation. This suggests that the risk of DSA development may not be fully explained by the quantitative epitope mismatch load alone.

Therefore, we hypothesize that certain epitopes may have higher immunogenic potential despite a lower overall mismatch load. We have included this point in the Discussion section: “Further studies are needed to explore the qualitative differences in epitope immunogenicity, as some epitopes may elicit stronger alloimmune responses than others.”

4. Minor revision: page 7 line 112: the term 'repeated DQ MM' is inappropriately used in my opinion, as it should instead refer to MMs which recurs in recipients of >1 graft.

Responses:

We appreciate your insightful comment regarding the term "repeated DQ MM." We agree that, in transplant immunology, this term typically refers to mismatches that recur in recipients of multiple grafts. However, in our manuscript, "repeated DQ MM" was used to describe cases where both donor HLA-DQ antigens mismatched the recipient with the same DQ allele (e.g., both donor HLA-DQ antigens are DQ5). To avoid confusion, we will revise the term to "duplicate DQ MM" to more accurately reflect the intended meaning and differentiate it from the standard usage of "repeated DQ MM" in the context of multiple grafts. We will update the manuscript accordingly for clarity.

From Reviewer #2,

In their manuscript "Variations in De Novo Donor-Specific Antibody Development Among HLA-DQ Mismatches in Kidney Transplant Recipients" Skulratanasak and colleagues retrospectively investigated a cohort of 491 kidney transplant recipients with HLA-DQB1 mismatches. They analyzed individually DQ loci carried by the recipient, and the likelihood of developing donor specific antibodies (DSAs). They found that patients who developed DSAs to HLA-DQ had increased risk of rejection and graft loss, in line with previously published results. They found that some recipient antigen mismatches such as DQ7 and DQ9 are at highest risk of inducing DSAs. The methodology used is mostly appropriate. The manuscript is sometimes difficult to follow. While there is some originality to the main finding of the manuscript, its clinical relevance needs to be further clarified. Specifically, it remains to be demonstrated whether the stratification in risk of developing DSAs (figure 3) also correlates with stratified rejection-free survivals and graft outcomes. I would encourage the authors to consider the following additional issues should they consider resubmitting the manuscript to a different journal.

Responses:

We sincerely thank the reviewer for the thorough evaluation of our manuscript and for providing constructive feedback. We appreciate the recognition of the originality of our study in identifying specific HLA-DQ mismatches, particularly DQ7 and DQ9, as high-risk for dnDSA development and the association with graft outcomes.

We greatly appreciate the reviewer’s encouragement to improve the clarity of our manuscript and will take this opportunity to strengthen both the scientific content and clinical implications of our findings. We hope that the revisions made in response to the reviewer’s suggestions will provide more clarity on the clinical relevance of our study.

1. Data on DR mismatches should be provided. In its current form, it is impossible to determine whether clinical outcomes were influenced by DR mismatches

Responses:

Thank you for your insightful comment regarding the potential influence of HLA-DR mismatches on clinical outcomes. We have now included data on HLA-DR mismatches in Table 2 of the revised manuscript to provide additional context.

In our Cox regression analysis evaluating the impact of HLA-DR mismatches on the development of anti-DQ dnDSAs, we found no significant association (HR: 1.27; 95% CI: 0.52–3.15, p = 0.6). This suggests that HLA-DR mismatches did not significantly influence the occurrence of anti-DQ dnDSAs in our cohort.

We appreciate your suggestion to include this additional information, which we believe enhances the completeness of our analysis and manuscript.

2. Prior studies have demonstrated that the number of DQ mismatches influences the development of DSAs. Here, the authors have focused on specific allele mismatches and development of DSAs. What was the distribution of single vs double mismatches in for each allele? In other words, were there allele mismatches that were more likely to be found in isolation and allele mismatches that were more likely to be found in a double mismatch? This could also influence differences in development of DSAs.

Responses:

Thank you for your insightful comment regarding the distribution of single vs. double MM for each HLA-DQ antigen and their potential influence on DSA development. We performed further analysis to address this question, and the detailed distribution of single and double MM for each specific HLA-DQ antigen has been provided in Supplement Table 1.

We found that the incidence of DSA development was 11.1% in recipients with a single MM and 17.3% in those with double MM (p = 0.124). While double MM appeared to carry a higher risk of DSA formation, likely due to an increased likelihood of antigen-induced immune responses, the rate of DSA development in double MM was not more than twice that of single MM. This suggests that while double MM may increase the risk, their impact on DSA development may not be solely proportional to the number of MM.

To improve clarity, we have updated the relevant statements in the manuscript as follows:

1. “Among them, 75 recipients had 2 HLA-DQB1 MM, bringing the total number of DQB1 mismatches in this study to 566. Details of the distribution of HLA-DQB1 MM are provided in Supplement Table 1.”

2. “The presence of dnDSAs against HLA-DQB1 was detected in 59 patients (12%) at a median of 4.23 years (IQR: 1.97–5.56) after KT. The mean MFI of the first maximum dnDSA was 9,274 ± 6,405. The incidence of HLA-DQ DSA development was 11.1% and 17.3% in patients with single and double MM, respectively.”

3. “The current concept regarding HLA and epitope mismatches focuses on the number of mismatches rather than the specific HLA types (30,31). Our study also revealed that recipients with double DQ MM tended to be at higher risk for DSA development than those with single MM (17.3% vs. 11.1%). However, the rate of DSA development in double MM was not twice as high as in single MM, suggesting that the quality of MM may play a more meaningful role in DSA formation than the quantity alone.”

We hope this additional analysis and the revised manuscript text adequately address your comment and improve the clarity and scientific depth of our study. Thank you again for your valuable suggestion.

3. How was AMR defined?

Responses:

Thank you for your comment. The diagnosis of AMR in our study was based on the Banff classification 2019. Pathological scoring, including glomerulitis (g) and peritubular capillaritis (ptc) scores, along with the presence of DSA and C4d staining, was reviewed to make the diagnosis. The revised text now reads:

"Diagnosis of AMR was determined by reviewing pathological scores, DSA results, and C4d staining in accordance with the Banff classification 2019 criteria."

We appreciate your suggestion, which has helped improve the clarity of our manuscript.

4. Was there a specific protocol for induction immunosuppression, or was this left to the discretion of the treating provider? Similarly, can you provide additional information on how maintenance immunosuppression was decided?

Responses:

Thank you for your comment regarding our induction and maintenance immunosuppression protocols. At our center, we follow a specific protocol for induction immunosuppression. IL-2 receptor antagonists are used for non-HLA-identical kidney transplant recipients, while antithymocyte globulin (ATG) is administered to high-risk recipients, such as those with high panel reactive antibody (PRA) levels. However, prior to 2012, most patients did not receive antibody induction due to limitations in reimbursement.

For maintenance immunosuppression, the initial regimen generally consisted of tacrolimus, mycophenolate mofetil (MPA), and prednisolone. In some cases, changes in immunosuppressive therapy were necessary due to clinical circumstances. For example, MPA was replaced with mTOR inhibitors in patients with BK virus viremia/viruria, persistent CMV viremia, mycobacterial infections, or those w

---

## [Decision Letter · Decision Letter 1]

10 Mar 2025

Variations in De Novo Donor-Specific Antibody Development Among HLA-DQ Mismatches in Kidney Transplant Recipients

PONE-D-24-21041R1

Dear Dr. Skulratanasak

We’re pleased to inform you that your manuscript has been judged scientifically suitable for publication and will be formally accepted for publication once it meets all outstanding technical requirements.

Kind regards,

Paolo Fiorina, MD, PhD

Academic Editor

PLOS ONE

Reviewers' comments:

Reviewer's Responses to Questions

**Comments to the Author**

1. If the authors have adequately addressed your comments raised in a previous round of review and you feel that this manuscript is now acceptable for publication, you may indicate that here to bypass the “Comments to the Author” section, enter your conflict of interest statement in the “Confidential to Editor” section, and submit your "Accept" recommendation.

Reviewer #2: (No Response)

Reviewer #3: All comments have been addressed

Reviewer #4: All comments have been addressed

Reviewer #5: All comments have been addressed

2. Is the manuscript technically sound, and do the data support the conclusions?

Reviewer #2: Yes

Reviewer #3: Yes

Reviewer #4: Yes

Reviewer #5: Yes

3. Has the statistical analysis been performed appropriately and rigorously? 

Reviewer #2: Yes

Reviewer #3: Yes

Reviewer #4: N/A

Reviewer #5: Yes

4. Have the authors made all data underlying the findings in their manuscript fully available?

Reviewer #2: Yes

Reviewer #3: Yes

Reviewer #4: Yes

Reviewer #5: Yes

5. Is the manuscript presented in an intelligible fashion and written in standard English?

Reviewer #2: Yes

Reviewer #3: Yes

Reviewer #4: Yes

Reviewer #5: Yes

6. Review Comments to the Author

Reviewer #2: (No Response)

Reviewer #3: (No Response)

Reviewer #4: (No Response)

Reviewer #5: The article Variations in De Novo Donor-Specific Antibody Development Among HLA-DQ

Mismatches in Kidney Transplant Recipients is very well tailored after extensive revision. Methods and results are well organized with coherent explanation and conclusion. This study is a significant addition to the literature on KT and the article can now be published.

7. PLOS authors have the option to publish the peer review history of their article (what does this mean? ). If published, this will include your full peer review and any attached files.

**Do you want your identity to be public for this peer review?** For information about this choice, including consent withdrawal, please see our Privacy Policy .

Reviewer #2: No

Reviewer #3: No

Reviewer #4: No

Reviewer #5: No

---

## [Editor Report · Acceptance letter]

PONE-D-24-21041R1

PLOS ONE

Dear Dr. Larpparisuth,

I'm pleased to inform you that your manuscript has been deemed suitable for publication in PLOS ONE. Congratulations! Your manuscript is now being handed over to our production team.

Kind regards,

on behalf of

Dr. Paolo Fiorina

Academic Editor

PLOS ONE